# Influence of Co-Parenting on Parental Burnout: A Moderated Mediation Model

**DOI:** 10.3390/bs14030159

**Published:** 2024-02-22

**Authors:** Qin Zhang, Mei Zhao

**Affiliations:** 1CAS Key Laboratory of Mental Health, Institute of Psychology, Chinese Academy of Sciences, Beijing 100101, China; zhangqin@psych-ac.cn; 2Department of Psychology, University of Chinese Academy of Sciences, Beijing 101408, China

**Keywords:** collaborative education, parenting burnout, family function, adolescents’ problematic behaviors, teenagers

## Abstract

Adolescence has always been regarded as a period of rapid psychological and behavioral change. Adolescents are subject to more difficult behaviors, and those difficult behaviors have a great impact on co-parenting and parenting burnout. In order to reveal the relationship between these factors, this study investigated the mediating effect of difficult adolescent behaviors on that relationship by constructing a theoretical model and examined the moderating effect of family functioning. In order to provide a scientific basis for preventing and intervening in adolescents’ problematic behaviors and improving parents’ parenting burnout, we conducted a study on the parents of 1638 teenagers in a junior high school in Huanggang City, China in May 2023, with a questionnaire filled out by the parents. The research tools included a parenting burnout questionnaire, Parental collaborative parenting Scale (PPCR), Adolescent Strengths and Difficult Behaviors Questionnaire (Parental Version), Family Function Scale, etc. An independent sample *t* test and ANOVA test were used to analyze whether there are certain demographic variables in parenting burnout, and SPSS27.0 was used for descriptive statistics, reliability and validity tests, correlation analysis and common method deviation tests. The adjusted mediation model was tested by using the SPSS macro program Process4.0. Results: The variance in the explanatory capacity of the largest factor in this study was 21.955%, which did not exceed the critical value of 40%, so there was no obvious common method deviation in the data of this study. The independent sample *t* test and ANOVA test showed that there are certain differences in parental rearing burnout dependent on parental gender, the main caregivers, family economic income and demographic variables. The results of the adjusted mediation model test by Process4.0 show the following: (1) Adolescent difficult behavior plays an intermediary role between parental collaborative parenting and parenting burnout; (2) the indirect effect of collaborative parenting on parenting burnout through adolescents’ problematic behaviors is regulated by family functions; (3) the relationship between adolescent difficult behavior and parenting burnout is regulated by family function; (4) the direct influence of collaborative parenting on parenting burnout is also regulated by family function. Conclusion: Adolescents’ difficult behavior partially mediates the influence of parents’ collaborative parenting on parenting burnout. In addition, family function not only mediates the front and back ends of mediation, but also mediates the direct influence of collaborative parenting on parenting burnout. These findings are instructive for improving family parenting problems and promoting adolescent development. The results of this study may be helpful in enhancing parents’ awareness of parenting of adolescents in China, which will provide reference for some teachers in China to understand adolescent behavior. At the same time, the results may provide new enlightenment for mental health professionals and enable them to fully understand the parenting contradictions between parents and adolescents in China.

## 1. Introduction

With the development of the social economy, the problem of parenting burnout has attracted more and more attention from academic circles. According to the international parenting burnout survey, parenting burnout is widespread around the world, ranging from about 1% to 8% [1,2]. Empirical research shows that parenting burnout is related to various adverse consequences. In parents, this may induce escape, insomnia, depression and reliance on corporal punishment, which will seriously affect the physical and mental health of parents and children. In addition, it will also hinder family relations and children’s growth [3,4].

In the past few decades, many studies have discussed the influence of family factors on parenting burnout. Among these family factors, the conflict, degradation and inconsistency of collaborative parenting should be one of the most important factors that lead to parents’ parenting burnout [5]. According to the family system theory, the conflict of collaborative parenting destroys predictability, stability and security, and causes confusion and hostility in the family environment, which may be harmful to the development of teenagers and lead to parenting burnout. However, previous empirical studies on the relationship between collaborative parenting and parenting burnout have given inconsistent results. Some studies show their close relationship [6]. Fretwell’s study concluded that conflictive parenting is more likely to lead to adolescents developing problematic behaviors such as substance abuse or addiction, which further contributes to parenting burnout [7]. Mikolajczak et al.’s study concluded that parents who use negative parenting behaviors such as neglect and domestic violence tend to cause adolescents to turn away from their parents, which in turn further increases the level of parenting burnout [8]. In contrast, Fretwell et al. concluded that parents who effectively control their emotions and enhance their perceptions of co-parenting are likely to reduce their levels of parenting burnout [9], while others show their weak or unimportant relationship. This inconsistent result may mean that some teenagers’ problems or difficult behaviors mediate the relationship between collaborative parenting conflict and parenting burnout. In the parental system, supportive collaborative parenting is a resource. It can improve the marital satisfaction between husband and wife and help them cope with risks and pressures effectively, thus reducing the possibility of parenting burnout [10]. On the contrary, conflicting or inconsistent collaborative parenting, as a stressor, intensifies parenting pressure, thus increasing the possibility of parenting burnout [11,12]. However, our understanding of the mechanism of how collaborative parenting affects parenting burnout is still relatively insufficient.

After reviewing a large number of related studies, we further concluded that family factors, especially family functions, can regulate the relationship between collaborative parenting conflicts and parenting burnout, and the direct impact of conflicting collaborative parenting on parenting burnout may be mediated by adolescents’ problematic behaviors. At present, the research on the antecedents of parenting burnout mainly focuses on social support and demographic variables [13]. It is found that when parents do not have enough resources to meet the needs of raising children, or when there is a persistent imbalance between parents’ parenting input and children’s behavioral performance, parents will experience burnout [14,15]. Although adolescent behavioral difficulties have been found to be an important reason for parents’ parenting burnout, family function has been proved to be a good way of alleviating parenting burnout [16].

Several theories and frameworks support our argument. For example, the theoretical framework proposed by Lui et al. [17] explains how the family and personal factors of teenagers interact and have an impact on parents’ parenting burnout. Yuan et al. [18] believe that adolescents’ difficult behavior plays an important role in the collaborative parenting conflict and parenting burnout. They further found that adolescents’ difficult behaviors and family function are widely regarded as the main individual factors affecting parenting burnout in the literature, and family function in family factors has exerted different influences and functions on collaborative parenting conflict, adolescents’ difficult behavior and parenting burnout. Therefore, collaborative parenting conflict may affect parenting burnout through adolescents’ difficult behavior [19,20].

Other theories also support or strengthen our proposition. According to the spillover hypothesis in the family system theory, collaborative parenting can affect the psychological and behavioral development of teenagers through other family subsystems [21]. As an important part of the parent–child system, collaborative parenting has a profound impact on teenagers’ problem behaviors. For example, the resource allocation model assumes that individuals’ cognitive resources are limited. Once teenagers encounter different challenges at the same time, they usually need to allocate cognitive resources to deal with each task, which may reduce their efficiency in completing the main tasks. Collaborative parenting conflict is one of the main influencing factors in teenagers’ lives, which will distract their attention and lead to some problems [22]. Under the cultural background of China, teenagers’ difficult behaviors, such as showing inattention or lack of effort and persistence in learning, are an important factor leading to psychological fatigue of parents in China [23]. Once teenagers encounter collaborative parenting conflicts in the family, their attention or participation in school activities may be reduced [24], which in turn has a negative impact on parents’ parenting burnout. In addition, the theory of emotional security holds that family conflicts will cause teenagers’ insecurity or negative emotional reactions, such as depression, destroy their healthy psychology, or increase their psychological pressure [25]. Teenagers who have experienced the conflict of collaborative parenting are more prone to emotional instability and will have more difficult behaviors. As far as we know, there is still a lack of empirical research on the indirect relationship between collaborative parenting and family function and parenting burnout through adolescent difficult behaviors. Therefore, it is still unknown how collaborative parenting, adolescent difficult behaviors and family function jointly affect parenting burnout. Given that China is a country with one of the highest female employment rates, with the liberalization of the three-child policy in China and the increase in the number of children, parents will face more responsibilities for education and care. The distribution of family resources, the establishment of the parent–child relationship and the maintenance of family harmony will become huge challenges [26]. Parents in dual-income households need to share the responsibility of care and participate in many aspects of parenting practice, such as caring for, educating and disciplining their children. However, under the cultural background of China, influenced by familism, parents in China often have different parenting philosophies and methods. These problems may lead to common parenting conflicts between parents. In this case, family functions and family atmosphere may be affected and changed, which may directly or indirectly hinder the development of teenagers [27,28,29]. However, existing studies have mostly focused on the structure and measurement of parenting burnout, influencing factors, theoretical models, and comprehensive studies on the causes and outcomes of parenting burnout using questionnaires, network analysis and other methods. In China, more research has been conducted on the review of parenting burnout and the localization of measurement tools, and not many empirical studies have been conducted on the influencing factors and internal mechanisms of parenting burnout. Meanwhile, the existing studies focus more on individual parents’ characteristics and personality factors, and seldom include variables such as family structure and children’s problematic behaviors.

In summary, the present study constructed a model to explore the relationship between parental co-parenting, adolescent difficult behaviors, family functioning, and parenting burnout. The hypotheses of this study were the following: (1) Parental co-parenting has a direct effect on parenting burnout; (2) adolescent difficult behaviors mediate the relationship between parental co-parenting and parenting burnout; (3) co-parenting is moderated by family functioning through adolescent difficult behaviors; (4) adolescent difficult behaviors are moderated by family functioning; and (5) the direct effect of co-parenting on parenting burnout is moderated by family functioning. The random sample of informational data for this study was obtained from parents of all students in grades 7–9 in a middle school in Huanggang City, Hubei Province, China. Figure 1 illustrates the moderated mediation model.

## 2. Materials and Methods

### 2.1. Data Acquisition and Research Object

The type of research design chosen was a correlational design, and the data collection method used a questionnaire survey, which was conducted between April 2023 and May 2023, and non-probability sampling to survey the parents of all of the students in grades 7–9 in a middle school in Huanggang City, Hubei Province. One week prior to the survey, parents were informed of the purpose and process of the questionnaire survey by the teachers at the school, and were told that the questionnaire should be filled out by the primary caregivers of the students. At the beginning of the survey, the class teacher distributed the Informed Consent Form, questionnaire link and QR code in the class WeChat group to allow parents to participate in the China Questionnaire Star online questionnaire survey. The confidentiality and anonymity of the participants’ relevant information was maintained throughout the survey. A total of 1638 questionnaires were collected, and a total of 976 valid questionnaires were obtained by eliminating samples that failed to meet the deadline for return, contained inconsistent responses to the validity check items, or were missing data.

The study was approved by the Ethics Committee of the Institute of Psychology, Chinese Academy of Sciences, and written informed consent was also obtained from the patients and their parents or legal guardians, as well as support and authorization from the principal of the participating middle school in Huanggang City, Hubei Province, China.

Table 1 shows the descriptive statistics of the participants. Among the 976 respondents, there were 9 respondents (0.9%) aged 20–30 years old, 519 respondents (53.2%) aged between 30 and 40 years old, and a total of 448 respondents (45.9%) aged 40 years old or older; 279 respondents (28.6%) were fathers, and 697 respondents (71.4%) were mothers; as the primary caregivers, 56 respondents (5.7%) were fathers, 364 respondents (37.3%) were mothers, 480 respondents (49.2%) were joint caregivers comprising husband and wife, and 76 respondents (7.8%) were other elders. The number of children per respondent household was convenient, with 118 households (12.1%) having only one child, 704 (72.1%) having two children, and 154 (15.8%) having three or more. In terms of work, those engaged in technical or mental labor accounted for 198 respondents (20.3%), and 473 respondents (48.5%) were engaged in manual labor, 147 (15.1%) were freelancers, and 158 (16.2%) were engaged in other work. In terms of family structure, there were a total of 599 (61.4%) ordinary families, 398 (31.6%) with extended family structure, 32 (3.3%) divorced families, and 11 (1.1%) reorganized families. From the point of view of the length of marriage, 22 respondents (2.3%) were married for less than 5 years, a total of 42 (4.3%) were married for 5–10 years, and the rest had been married for more than 10 years (93.4%). From the point of view of the place of residence, 304 respondents (31.1%) lived in rural villages, 641 (65.7%) lived in towns, and 31 (3.2%) lived in the city.

### 2.2. Research Tools

#### 2.2.1. Parental Collaborative Parenting Questionnaire

This study used the Chinese version of Machle’s Collaborative Parenting Questionnaire (Parents Co-Parenting Rating Scale for Assessment Co-Parenting) to measure collaborative parenting [30]. This variable is measured through four measurement dimensions. These four measurement dimensions are divided into unity, consistency, conflict and degradation. Each measurement dimension contains 6–8 measurement questions, including 29 questions in total. The questionnaire is scored with a 7-point Likert scale, and the integral range is from “never” to “always” with 1 to 7 points, respectively. Among them, the main items of the unity dimension include “I will affirm or praise my child in front of my lover” (factor loading = 0.969), “I will promote the happy interaction between my lover and my child” (factor loading = 0.972) and “I will express my feelings to my child with body language (such as hugging and touching)”. The main items of the consistency dimension include “I will take the initiative to discuss with my lover in restraining children’s behavior” (factor loading = 0.941), “When children do something wrong, my lover and I handle it in the same way” (factor loading = 0.774) and “When I reward children, I will use the method approved by my lover” (factor loading = 0.787). The main items of the conflict dimension include: “I will say things that hurt my lover” (factor loading = 0.834), “I am hostile to my lover” (factor loading = 0.865) and “I will argue with my lover” (factor loading = 0.903); The main items that belittle the dimension include: “I will express my feelings to my lover with body language (such as hugging and touching)” (factor loading = 0.978),”I will talk to my child about my loved one (your mom/dad) doing things incorrectly (factor loading = 0.897)”, “When interacting with my child, I will talk about my loved one’s shortcomings” (factor loading = 0.973). In this study, the Cronbrch’sa SA coefficient of the whole scale was 0.933, and the α coefficients of each dimension of the scale were 0.885, 0.914, 0.886 and 0.918, respectively. The scale has good reliability and can be used as a research tool for the collaborative parenting of parents in China.

#### 2.2.2. Parenting Burnout Scale

This variable was measured by a scale-based questionnaire, which was verified by China scholar Li Yongxin [31] and others in 2021. It includes 7 questions, with 7 points for scoring, where 1 means “never” and 7 means “every day”. The higher the score, the higher the parenting burnout is. The seven items are “I can’t stand the identity of parents any more” (factor loading = 0.917), “I don’t want to take on the role of parents again” (factor loading = 0.856) and “I wake up in the morning and think of taking care of my children all day. I feel very tired (even if I haven’t faced my children yet)“ (factor loading = 0.930), "I feel I can’t do the role of a parent“ (factor loading = 0.861), ”I don’t think I am a good parent as before“ (factor loading = 0.893), “I can’t be a good parent anymore” (factor loading = 0.949), “I want to express my love to my children as I used to, but I feel powerless” (factor loading = 0.964). The internal consistency coefficient α of the scale in this study was 0.906, and the reliability of the scale was good.

#### 2.2.3. Adolescent Strengths and Difficulties Questionnaire

We used the Chinese version of the Strengths and Difficulties Questionnaire (SDQ) to measure adolescents’ problem behaviors. The SDQ was developed by American psychologist R. Goodman in 1997 and revised by China scholar Kou Jianhua [32] and others in 2005. The SDQ is a tool to evaluate adolescents’ behavior and emotional problems. It has good reliability and validity.

This questionnaire uses a Likert three-point scoring method, with 0–2 representing “nonconformity”, “somewhat conformity” and “very conformity”, respectively. The main items of this questionnaire are “often losing temper or making a scene” (factor loading = 0.754), “often fidgeting or fidgeting” (factor loading = 0.754) and “easily distracted, Inattention” (factor loading = 0.754), “under new circumstances, people who are nervous or cling to adults are prone to lose confidence” (factor loading = 0.754), “lying or cheating often” (factor loading = 0.754) and “getting along with adults is more harmonious than getting along with children” (factor loading = 0.754). In addition, the reliability of the peer relationship question in this questionnaire was low, so it was deleted in this study, and the conduct question and hyperactivity attention could be combined into externalized questions, and the total score of difficulty was composed of externalized questions and emotional symptoms. In this study, the internal consistency coefficient α of externalized questions was 0.820, the internal consistency coefficient α of emotional symptoms was 0.889, and the consistency coefficient α of the total score of difficult questions was 0.879. Overall, the reliability of the scale was good.

#### 2.2.4. Family Function Scale

Family function was measured by the family function scale, which adopted the Familial Aptitude and Adaptation Scale (Faces) compiled by Olson et al. and was revised by Fei Lipeng [33]. Faces include two subscales: one’s actual cognition and ideal cognition of family, and the contents of the two scales are consistent, comprising 30 questions each. Considering the convenience of answering and the practical significance of the test results, we only used a sub-scale to measure a person’s actual perception of his family. This scale includes the two dimensions of family intimacy and family adaptability, which refer to the emotional connection between family members, and comprises 16 questions. The main items include “Family members will try their best to support each other when there are difficulties” (factor loading = 0.936), “All family members get together for activities” (factor loading = 0.864), “At home, Everyone should do things together“ (factor loading = 0.824), ”family members are familiar with each member’s close friends“ (factor loading = 0.931), ”When family members want to make decisions, I like to discuss with my family“ (factor loading = 0.862), ”In our family, all the entertainment activities are done by the whole family“ (factor loading = 0.941) and ”In our family, all the entertainment activities are done by the whole family “(factor loading = 0.863). Family adaptability refers to the ability of the family system to change with the family situation and problems in different stages of family development. For 14 questions, this scale only uses the sub-scale of the actual feelings about the current situation of one’s own family. The scale uses a Likert 5-point score, where 1 means “no” and 5 means “always”. The higher the score of each dimension, the better the function of this dimension. The higher the total score of the scale, the better the family function. The main items include “In our family, every member can express his opinions at will” (factor loading = 0.782), “every family member participates in making important family decisions” (factor loading = 0.956), “Teaching the elders, The younger generation can express their opinions“ (factor loading = 0.832), ”family members discuss problems together and are satisfied with the solution of the problems“ (factor loading = 0.754), ”in the family, We take turns to share different housework“ (factor loading = 0.864), ”When the family situation changes, it is easy to make corresponding changes in the family’s ordinary life rules and family rules“ (factor loading = 0.903), ”When there is a contradiction in the family, members give in to each other and compromise“ (factor loading = 0.750), ”Can children’s suggestions be accepted when solving problems? “ In this study, the consistency coefficient of the scale was 0.936, in which the Cronbrch’sa coefficient of family intimacy dimension was 0.873, and the Cronbrch’sa coefficient of family adaptability dimension was 0.881. The Chinese version of the Faces questionnaire has good reliability and can be used as a tool for family function research.

### 2.3. Statistical Analysis

SPSS27.0 was used for descriptive statistics, reliability and validity tests, correlation analysis and common method deviation test. The adjusted mediation model was tested by using the SPSS macro program Process4.0 plug-in.

## 3. Results

### 3.1. Common Method Deviation Test

Because the data of this study were all from the same subjects’ reports, there may be have been common method deviation. In view of this, this study monitored the data quality from two aspects: program control and statistical control. In the program control, the importance of this study was conveyed to teachers before the research was carried out, and the confidentiality of the research data and the follow-up research were explained in the questionnaire guidance part to improve the concentration of the subjects in the process of filling out the questionnaire. In the statistical control, the single-factor Harman test method was used, and the principal component analysis method was used to extract the factors from all of the variables in the questionnaire. Finally, 19 factors with eigenvalues greater than 1 were obtained, of which the variance of the largest factor was 21.955%, which did not exceed the critical value of 40%. Therefore, it is considered that there was no obvious common method deviation in the data of this study, which met the premise requirements of subsequent statistical tests.

### 3.2. Mean, Standard Deviation and Correlation Analysis of Each Variable

The correlation matrix of each variable is shown in the following Table 2. In the independent sample *t* test and ANOVA test, it was found that there were differences in certain demographic variables in parental rearing burnout. Specifically, there were significant differences between father and mother in parenting burnout. As for the main caregivers, the parenting burnout of families with parents as co-caregivers was significantly lower than that of families with parents as sole caregivers. In terms of family economic income, the parenting burnout of families with average family income was significantly lower than that of families with below-average family income. In addition, the correlation analysis showed that collaborative parenting was negatively correlated with adolescents’ difficult behavior and parental parenting burnout, and positively correlated with family function, indicating that in families with better collaborative parenting, children’s difficult behaviors were low, and parents’ parenting burnout was also low. In addition, adolescents’ difficult behavior was positively correlated with parenting burnout and negatively correlated with family function. Family function was negatively correlated with parenting burnout.

### 3.3. The Relationship between Collaborative Parenting and Parenting Burnout: A Test of the Mediated Model

According to Muller, Judd and Yzerbyt [34], and Wen and Ye [35], it is necessary to estimate the parameters of three regression equations to test the regulated mediation model. Equation (1) [34] estimates the moderating effect of the moderating variable (family function) on the relationship between the independent variable (collaborative parenting) and dependent variable (parenting burnout); Equation (2) [35] estimates the moderating effect of the moderating variable (family function) on the relationship between the independent variable (collaborative parenting) and intermediary variable (adolescent strengths and difficult behaviors); Equation (3) [35] estimates the moderating effect of the moderating variable (family function) on the relationship between the intermediary variable (adolescents’ strengths and difficult behaviors) and dependent variable (parenting burnout) and the moderating effect of the independent variable (collaborative parenting) on the residual effect of the dependent variable (parenting burnout). All of the predictive variables are standardized in each equation, and the variables such as parental gender, main caregivers and family income are controlled (in the difference test, the three variables of parental gender, main caregivers and family income have statistically significant effects on the dependent variables). The variance expansion factor of all predictive variables was not higher than 1.37, so there was no multicollinearity problem.

If the model estimates that it meets the following two conditions, it shows that there is a mediated effect: (a) In Equation (1), the total effect of collaborative parenting is significant, and the size of this effect does not depend on family function; (b) In Equation (2) and Equation (3), collaborative parenting has a significant effect on adolescents’ strengths and difficult behaviors, while the interaction between adolescents’ difficult behaviors and collaborative parenting has a significant effect on parenting burnout, and/or the interaction between collaborative parenting and family function has a significant effect on adolescents’ strengths and difficult behaviors.

As shown in Table 3, Equation (1) is significant as a whole. Collaborative parenting negatively predicts parenting burnout, and the interaction between collaborative parenting and family function has a significant predictive effect on parenting burnout. Equation (2) is significant on the whole, in which collaborative parenting negatively predicts adolescents’ strengths and difficult behaviors, the interaction between collaborative parenting and family function also significantly predicts adolescents’ strengths and difficult behaviors, and the second half of the mediation effect, that is, adolescents’ problematic behaviors, also significantly predicts parenting burnout. Finally, Equation (3) is significant on the whole, in which adolescents’ strengths and difficult behaviors positively predict parenting burnout, the first half of the mediation effect, namely collaborative parenting, significantly predicts adolescents’ problematic behaviors, and the interaction between adolescents’ problem behaviors and family functions can predict parenting burnout.

In order to reveal the essence of this interaction effect more clearly, we calculated the effect value of collaborative parenting on parenting burnout in Equation (1) when the family function was the average plus or minus one standard deviation (that is, the single slope test), the effect value of collaborative parenting on adolescents’ strengths and difficult behaviors in Equation (2) (that is, simple slope test), and the effect value of adolescents’ strengths and difficult behaviors on parenting burnout in Equation (3). According to the regression equation, the simple effect analysis diagram was drawn by taking the values of the independent variable of the corresponding equation and the average family function plus or minus one standard deviation (see Figure 1 and Figure 2 for details). The test in Figure 1 shows that collaborative parenting has a significant negative predictive effect on parenting burnout when the family function level is low (Bsimple = − 0.09, SE = 0.05, *p* = 0.048). When the family function level is high, collaborative parenting has a significant negative predictive effect on parenting burnout (Bsimple = − 0.29, SE = 0.05, *p* < 0.001).

Therefore, the interaction model in Equation (1) conforms to the promotion hypothesis of “protection factor-protection factor model” rather than the exclusion hypothesis. The test in Figure 2 shows that collaborative parenting has a significant negative predictive effect on adolescents’ difficult behavior when the family function level is low (Bsimple = − 0.26, SE = 0.04, *p* < 0.001). When the family function level is high, collaborative parenting has a significant negative predictive effect on adolescents’ problematic behaviors (Bsimple = −0.36, SE = 0.05, *p* < 0.001). Therefore, the interaction model in Equation (2) conforms to the promotion hypothesis of “protective factor-protective factor model” rather than the exclusion hypothesis. The test in Figure 2 shows that when the family function level is low, adolescents’ problematic behaviors contribute to parenting burnout (Bsimple = 0.30, SE = 0.05, *p* < 0.001). When the level of family function is high, adolescents’ difficult behaviors have a significant positive predictive effect on parenting burnout (Bsimple = 0.03, SE = 0.05, *p* = 0.496), so the interaction model in Equation (3) conforms to the exclusion hypothesis of “protective factor-protective factor model” rather than the promotion hypothesis.

Generally speaking, we found that the influence of collaborative parenting on parenting burnout is mediated by adolescents’ strengths and difficult behaviors, and regulated by family functions; family function can adjust the direct influence of collaborative parenting on parenting burnout. For families with high family function, collaborative parenting has a strong direct effect on parenting burnout, and it has a strong indirect effect on the first half of parenting burnout through adolescent difficult behavior. For families with low family function, it has a weak direct effect on parenting burnout, and it has a weak indirect effect on the second half of parenting burnout through adolescent difficult behavior.

## 4. Discussion

This study constructed a mediating model to study the mediating role of adolescents’ strengths and difficult behaviors in the relationship between collaborative parenting and parenting burnout, and the mediating role of family function in that relationship. The results of this study are helpful to understand how collaborative parenting affects parenting burnout and the conditions under which it occurs. The results of this study have theoretical and practical significance for future research in this field and intervention measures against parenting burnout.

### 4.1. The Correlation between Collaborative Parenting and Parenting Burnout

Zhou et al. [36] analyzed data from 283 pairs of parents in China using a subject–object reciprocity model in 2023 and found that co-parenting by each parent significantly and negatively predicted their respective parenting burnout, while Qiao et al. [37] found through a questionnaire survey of mothers of 1164 children in Guangdong Province, China (Mage = 4.26 ± 0.85) that co-parenting significantly and negatively affected parenting burnout. Equation (1) in the present study was significant overall, indicating that co-parenting negatively predicted parenting burnout; therefore, the findings of these two studies were consistent. Parents are more likely to be involved in their children’s parenting process when they hold a consensus view on parenting. This agreement creates a greater sense of well-being and fulfillment, which in turn reduces the level of stress in the parenting process. In contrast, parents face more resistance and stress from their partner if there is conflict, criticism, or devaluation of the other person in the parenting process. This situation leads to parenting burnout. According to the risk–resource balance theory, parenting burnout occurs when parents are exposed to excessive stress without access to sufficient resources to alleviate it. Nunes et al. [10]. argued that supportive co-parenting in the parenting system serves as a resource that can enhance marital satisfaction between couples and help them cope with risk and stress effectively, thus contributing to the reduction of parenting burnout, which is the same as the conclusion reached in the present study, which will provide a useful direction for the design of future parenting burnout intervention programs.

### 4.2. The Intermediary Role of Adolescents’ Strengths and Difficult Behaviors

Currently, international research on the mediating role of difficult adolescent behaviors between co-parenting and parenting burnout is extremely rare, and mainly focuses on the study of two of these variables. Liu and Li [38] followed up 157 preschool children for a period of one year, with the children’s mothers completing the Perceptions of Co-parenting Questionnaire, the Strengths and Difficulties of Children Questionnaire, the Marital Conflict Scale, and the Parenting Relationship Scale. The results showed that there was a significant correlation between co-parenting and children’s behavioral problems, and that co-parenting indirectly affects children’s behavioral problems through marital conflict in addition to directly affecting children’s behavioral problems. The results of this study share commonalities with the results of our study, which showed that there was a significant correlation between co-parenting and children’s behavioral problems. Liu and Li believe that co-parenting indirectly affects children’s behavioral problems through marital conflict in addition to directly affecting children’s behavioral problems, according to the results of their study. Conflict indirectly affects children’s behavioral problems, and according to the Parenting Co-Parenting Scale, marital conflict is also a part of co-parenting; therefore, this still essentially proves the findings that there is a significant correlation between co-parenting and children’s behavioral problems. Our findings are theoretically supported by family systems theory, which states that supportive or cohesive parenting provides children with a loving atmosphere and an environment for observational learning, which facilitates the learning and internalization of positive behaviors, according to the spillover hypothesis of family systems theory. In contrast, negative co-parenting perpetuates negative emotional environments, leading to the emergence of a variety of problem behaviors. These problematic behaviors, in turn, influenced co-parenting behaviors and increased parenting stress, which in turn led to parenting burnout, suggesting that adolescent difficult behaviors mediate the relationship between co-parenting and parenting burnout, but the extent to which adolescent difficult behaviors mediate the relationship between co-parenting and parenting burnout remains underexplored. Our findings suggest that collaborative parenting significantly predicts adolescent difficult behavior, which in turn influences parenting burnout. Thus, adolescents’ problem behaviors mediate the relationship between co-parenting and parenting burnout. Our results also provide evidence of the adverse effects of negative co-parenting relationships on adolescents; negative co-parenting tends to trigger more problems in adolescents. Li [39] argued that adolescents’ personality and outward appearance play a role in the relationship between family functioning and parenting burnout. In addition, they emphasize the interactions between the individual and the environment, and the results of our study are again similar to Li’s view to a certain degree. This also supports the mediating role of adolescents’ difficult behaviors.

### 4.3. The Regulatory Role of Family Functions

This study found that family functioning moderated not only the direct effects of co-parenting on parenting burnout, but also the effects of co-parenting on adolescents’ difficult behaviors and the effects of adolescents’ difficult behaviors on parenting burnout. This study found that in the direct relationship between co-parenting and parenting burnout, co-parenting had a greater protective effect on parenting burnout in families with higher levels of family functioning, a finding that is similar to the results of a study by Yuan et al. [18], which also showed that family functioning moderated the relationship between parenting burnout and middle school students’ problem behaviors. The difference is that the dependent variable is different, and both studies reflect the important moderating role of family functioning between the two. Vinces-Cua [40] investigated the moderating role of family functioning by establishing a relationship between parenting and adolescent difficult behaviors, and the results of his study showed that focusing on improving family relationships or family functioning helps to moderate or improve parental co-parenting and has a positive impact on changing adolescent difficult behaviors. Although this study did not explicitly state the moderating role of family functioning on both co-parenting and adolescent difficult behaviors, there are commonalities between his findings and some of our findings. We both agree that the protective effect of co-parenting on adolescent difficult behaviors is stronger in families with higher levels of family functioning. Regarding the relationship between adolescent difficult behaviors and parenting burnout, individuals whose children exhibit problem behaviors are less likely to experience parenting burnout if they have higher levels of family functioning. Combined with these theories, this study concludes that good family function could effectively improve negative parenting behavior. Especially in families with negative collaborative parenting dynamics, good family function helps parents adapt to conflicts and emotional resistance more quickly, and enables them to obtain protective resources, thus reducing parenting burnout. Similarly, children are more likely to feel warm in the family and are less likely to have problem behaviors. In families with problem behaviors, good family function enables children to have positive role models and parents’ love, gradually correct their behaviors, and alleviate parents’ parenting burnout.

### 4.4. Limitations and Future Research Directions

Although this study has certain value, there are still some areas to be improved. First, this study used a questionnaire survey to collect data, and parents filled out the questionnaire about adolescents’ problematic behaviors. In the future, different methods can be used to collect data, such as teacher reports or teenagers’ self-reports, to avoid the influence of social approval and common method variation. Secondly, this study adopted cross-sectional research. Whether the research results are reliable or not needs to be further investigated by longitudinal research, which has causal inference. Therefore, future research should be conducted in a longitudinal way to establish the causal relationships between variables. Finally, our sample was composed of parents of students in grades 7 to 9. Further research is needed to test the universality of our findings in a wider population.

## 5. Conclusions

The results of this study show that collaborative parenting is negatively related to parenting burnout, adolescent strengths and difficult behaviors, and positively related to family function. Adolescents’ strengths and difficult behaviors are positively related to parenting burnout, and family function is negatively related to parenting burnout. After controlling parental gender, main caregivers and family income, collaborative parenting directly predicts parenting burnout and indirectly affects parenting burnout through adolescents’ problem behaviors. Family function regulates the direct and indirect effects of collaborative parenting on parenting burnout.

Generally speaking, the research results emphasize the importance of considering the interaction between specific family backgrounds and different backgrounds for scientifically understanding parenting burnout and formulating effective intervention measures. We suggest that improving family functions, improving perceptions of collaborative parenting with adolescents, correctly understanding difficult adolescent behaviors, creating a harmonious family atmosphere and promoting positive interaction among family members will help prevent and alleviate parents’ job burnout.

## Figures and Tables

**Figure 1 behavsci-14-00159-f001:**
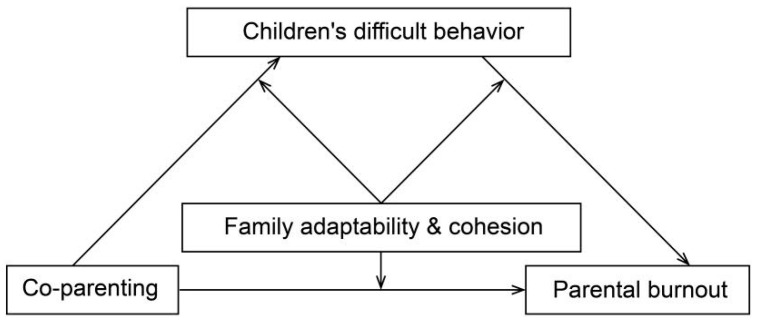
Image of mediation model with adjustment.

**Figure 2 behavsci-14-00159-f002:**
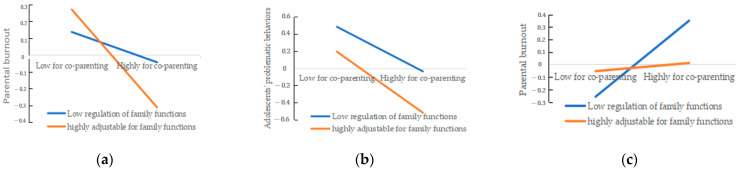
Response surface of interaction factors to specific power consumption. (**a**) The moderating effect of family functioning on the relationship between co-parenting and parenting burnout; (**b**) the moderating effect of family functioning on the relationship between co-parenting and children’s difficult or problematic behaviors; (**c**) the moderating effect of family functioning on the relationship between children’s difficult or problematic behaviors and parenting burnout.

**Table 1 behavsci-14-00159-t001:** Descriptive statistics of participants.

Variables		n	%	Variables		n	%
Age	20–30	9	0.9	Number of children	1	118	12.1
	30–40	519	53.2		2	704	72.1
	Above 40	448	45.9		3 and above	154	15.8
Gender	Father	279	28.6	Type of work	Technical staff	198	20.3
	Mother	697	71.4		Manual laborers	473	48.5
Main Dependent	Father	56	5.7		Freelance	147	15.1
	Mother	364	37.3		Others	158	16.2
	Co-parenting	480	49.2	Place of residence	Village	304	31.1
	Other elders	76	7.8		Town	641	65.7
Education Level	Junior high school and below	572	58.6		City	31	3.2
	High school	281	28.8	Length of marriage	Less than 5 years	22	2.3
	College	84	8.6		5–10 years	42	4.3
	Undergraduate and above	39	4.0		More than 10 years	912	93.4
Total		976	100.0	Total		976	100.0

**Table 2 behavsci-14-00159-t002:** Mean value, standard deviation and correlation of each variable.

Variable	M	SD	1	2	3	4	5	6	7
1 Parental gender	1.71	0.45	one						
2 Dependent	2.59	0.716	−0.00	one					
3 Income	3.35	0.65	−0.02	−0.09 **	one				
4 PPCR	5.21	0.85	−0.077 *	0.10 **	−0.18 ***	one			
5 SDQ	8.19	4.42	−0.02	−0.05	0.15 ***	−0.43 ***	one		
6 FACES	115.22	18.08	−0.03	0.083 **	−0.14 ***	0.66 ***	−0.39 ***	one	
7 Pb	1.57	0.94	0.06	−0.12 ***	0.17 ***	−0.29 ***	0.28 ***	−0.23 ***	1

Note: n = 976, M is the average, Income is the monthly household income; Pb is the parenting burnout, and the correlation coefficient is obtained by Bootstrap method. * means *p* < 0.05, ** means *p* < 0.01, and *** means *p* < 0.001, the same below.

**Table 3 behavsci-14-00159-t003:** Test on the mediating effect of cooperative education on parenting burnout.

Predictor Variable	Equation (1) (Dependent Variable: Parenting Burnout)					Equation (2) (Dependent Variable: Adolescent Problem Behavior)					Equation (3) (Dependent Variable: Parenting Burnout)				
	B	SE	P	95%CI		B	SE	P	95%CI		B	SE	P	95%CI	
Parental gender	0.11	0.06	0.09	−0.02	0.24	−0.10	0.06	0.10	−0.22	0.02	0.11	0.06	0.09	−0.02	0.24
Main supporter	−0.11	0.04	0.00	−0.20	−0.04	0.01	0.04	0.88	−0.07	0.08	−0.11	0.04	0.00	−0.20	−0.04
Family economic income	0.15	0.04	<0.001 ***	0.06	0.25	0.10	0.04	0.01	0.02	0.2	0.15	0.04	<0.001	0.06	0.25
X	−0.19	0.04	<0.001 ***	−0.27	−0.11	−0.31	0.04	<0.001	−0.38	−0.23	−0.19	0.04	<0.001 ***	−0.27	−0.11
X × W	−0.10	0.03	<0.001 ***	−0.15	−0.05	−0.04	0.02	0.03 *	−0.09	−0.01					
M						0.17	0.03	<0.001 ***	0.10	0.23					
M × W											−0.14	0.03	<0.001 ***	−0.20	−0.07

Note: X: collaborative parenting, W: family function, M: adolescent difficult behavior. * *p* < 0.05, *** *p* < 0.001, The 95% confidence interval of each predictive variable was obtained by Bootstrap method.

## Data Availability

The datasets used and/or analyzed during the current study are available from the corresponding author upon reasonable request.

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
