# Peer review of "Influence of Co-Parenting on Parental Burnout: A Moderated Mediation Model"

_behavsci, 2024, doi:10.3390/bs14030159_

Round 1

Reviewer 1 Report

Comments and Suggestions for Authors

It would be nice if the abstract includes a brief discussion on the practical implications of the study's findings for parents, educators, and mental health professionals.

The introduction provides a comprehensive overview of the problem of parenting burnout and outlines the rationale for the study. However, there are areas where clarity and precision can be improved, and some suggestions for enhancement are provided below.

Clarity and Conciseness:

The introduction contains numerous lengthy sentences that may hinder readability. Consider breaking down complex sentences for better clarity and understanding.

Empirical Research Gap:

The introduction outlines the background of the research, highlighting the global prevalence of parenting burnout and its consequences. However, it lacks a clear statement regarding the existing research gap or specific shortcomings in the literature that the current study aims to address.

Inconsistency in Terminology:

The terms "cooperative parenting" and "collaborative parenting" are used interchangeably. To maintain consistency, choose one term and use it consistently throughout the introduction.

Empirical Studies Review:

The introduction mentions previous empirical studies on the relationship between collaborative parenting and parenting burnout but does not provide a summary of their findings. Including a brief synthesis of existing research would strengthen the foundation for the current study.

Research Questions or Hypotheses:

The introduction lacks explicit research questions or hypotheses that the study aims to address. Clearly stating these elements would enhance the reader's understanding of the study's objectives.

The method section is generally well-structured and detailed. However, it would be improved:

Data Acquisition:

The section provides a clear description of the data acquisition process, including obtaining written consent and the support of the school principal. However, it would be beneficial to include information on how confidentiality and anonymity of participants were ensured to address ethical considerations.

Sample Size and Demographics:

The sample size is clearly mentioned (1,638), but the rationale behind this specific size is not explained. Including a brief justification or power analysis would enhance the methodological transparency. Additionally, the demographic information about fathers and mothers in the sample is useful.

Collaborative Parenting Questionnaire:

The detailed breakdown of dimensions and specific items in the collaborative parenting questionnaire is well-presented. However, some explanations appear to be repeated (e.g., item descriptions), and streamlining these details could enhance clarity.

The results section is generally well-executed, providing a detailed analysis of the data.

The discussion section is well-structured and provides a comprehensive analysis of the study's results.

Suggestions for Improvement:

·         Consider breaking down complex sentences for better readability.

·         Explicitly state the contributions of the study to the field.

·         Provide concise summaries or synthesis of key findings at the end of each subsection.

·         Include effect sizes or practical implications to enhance the interpretation.

·         Elaborate on how overcoming limitations could strengthen the study.

Reviewer 2 Report

Comments and Suggestions for Authors

Abstract: There is a missing part concerning the introduction, do not start directly with the objective.

Introduction: The introduction is very well supported by recent and recognized international studies. Even so, at the end the objectives should be more clearly stated, the generic one and then the specific ones based on the results presented later.

Method: It is not well described what type of research it is.The sample and the research procedure should be described as separate sections. Specifically, regarding the sample, the type of sampling, the sample inclusion criteria and the statistical parameters of the sample size are not determined. Regarding the procedure, everything related to the ethical parameters is missing, identifying the code of ethics committee. 

Data analysis: Many data regarding normality test, applied statistics, significance values etc. are missing.

Results: Perfect. 

Discussion: A discussion cannot be made using only a few studies, it is necessary to argue that part with the background so that a real construction of knowledge is made between what was investigated and the findings obtained. 

Conclusions: A part of implications closer to the educational and family field is missing.

Bibliography: More bibliographies from countries outside Asia

Comments on the Quality of English Language

Any comments

Round 2

Reviewer 2 Report

Comments and Suggestions for Authors

There are still some aspects of the submitted manuscript that need to be corrected: 

1. it does not present the type of the research design should go before the sample described. 

2. It should be specified that it is a non-probabilistic sampling.

3. The section of the procedure is not described in which it is indicated how the questionnaire was administered, communication processes with the center, etc. 

4. The section on data analysis is still obsolete.

5. The discussion section cannot be with 4 references; a true discussion of the results obtained with the findings of previous studies must be carried out, that is what a discussion is about. In this case, in the article, a summary of the results is made and that is not the objective of the discussion section, on the contrary, it is about contrasting the results with the previous scientific background. 
